# Measurement properties of the sit-to-stand test in people with chronic obstructive pulmonary disease: Protocol for a systematic review and meta-analysis using the COSMIN guidelines

**Christopher Farley**[1], **Stuart M. Phillips**[2], **Jenna Smith-Turchyn**[1], **Dina Brooks**[1,3]*

1 School of Rehabilitation Science, Faculty of Health Science, McMaster University, Hamilton, Ontario, Canada, 2 Department of Kinesiology, McMaster University, Hamilton, Ontario, Canada, 3 Department of Respiratory Medicine, West Park Healthcare Centre, Toronto, Ontario, Canada

☯ These authors contributed equally to this work.
* brookd8@mcmaster.ca

**Data Availability Statement:** No datasets were generated or analysed during the current study. All

## Abstract

### Introduction

Characteristics of chronic obstructive pulmonary disease (COPD) can include shortness of breath, chronic cough, sputum production and reduced exercise capacity. The sit-to-stand (STS) test variations (e.g., 5-repetition STS, 30-second STS) may be appropriate outcome measures to assess exercise capacity in people with COPD. To date, the measurement properties of the various STS tests in people with COPD have not been synthesized in a systematic review since the publication of the COnsensus-based Standards for the selection of health Measurement Instruments (COSMIN) guidelines. The purpose of this proposed systematic review is to synthesize the literature of the measurement properties of the STS test variations among people with COPD.

### Materials and methods

The review will be conducted with methods consistent with the COSMIN guidelines. Peer-reviewed publications will be included if they assessed the measurement properties (reliability, validity, responsiveness) of a STS test in community-dwelling adults with COPD. We will search six databases from inception. Study selection and data extraction will be conducted independently and in duplicate. We will assess the risk of bias using the COSMIN Risk of Bias tool, assess results against the COSMIN updated criteria for good measurement properties, and summarize certainty of evidence using the modified Grading of Recommendations Assessment, Development and Evaluation (GRADE) tool. Study results will be pooled by calculating weighted means and 95% confidence intervals or summarized narratively, as appropriate.

relevant data from this study will be made available upon study completion.

**Funding:** CF received funding from the Ontario Graduate Scholarship program. The funders had no role in study design, data collection and analysis, decision to publish, or preparation of the manuscript.

**Competing interests:** The authors have declared that no competing interests exist.

## Conclusion

This will be the first systematic review to synthesize the measurement properties of the STS tests in people with COPD using the methods recommended by COSMIN. Consequently, its results will be robust and may help clinicians or researchers identify the best variation of the STS test to use in their practice.

## Introduction

The Global Initiative for Chronic Obstructive Lung Disease (GOLD) defines chronic obstructive pulmonary disease (COPD) as a progressive and heterogenous lung condition which is characterized by dyspnea, chronic cough, sputum production and recurrent lower respiratory tract infections [1]. A systematic review and modelling analysis estimated that 392 million people aged 30 to 79 years met GOLD diagnostic criteria for COPD globally in 2019 [2]. More recently, a modelling study estimated the global prevalence of COPD in people over 25 years to reach 592 million by 2050 [3], highlighting the growing burden of this condition.

Pulmonary rehabilitation is a multidisciplinary intervention designed to improve and promote the long-term physical and psychological wellbeing of people with chronic respiratory disease [4]. It is a cornerstone treatment for people with COPD [5] due to its capacity to reduce dyspnea and fatigue symptoms and promote improved quality of life and exercise capacity [6]. Components of pulmonary rehabilitation include but are not limited to, exercise, education and behaviour change treatments [4]. Due to the exercise intolerance experienced by people with COPD [1], there is a need for an outcome measure to easily assess changes in exercise capacity in this population.

The 6-minute walk test (6MWT) is a widely used outcome measure to assess exercise capacity in people with chronic respiratory disease, including those with COPD [7]; however, the 6MWT may be too demanding for those who are severely impaired [7] and it requires a corridor of at least 30 meters in length which may not be available, especially in the community. An alternative outcome measure is the sit-to-stand (STS) test [8]. The sit-to-stand test was originally developed as a standardized measure to assess lower extremity strength by recording the time a person requires to complete 10 STS repetitions [9]. In contrast to the 6MWT, the STS test requires minimal space, which makes community implementation more feasible. There are several variations of the STS, which vary by the duration of the assessment or the number of repetitions (e.g., 30-second STS, 1-minute STS, 5-repetition STS) [10]. With multiple STS variations available, identifying the optimal version to utilize is challenging. A 2017 literature review aimed to summarize the psychometric properties of the STS test variations [11]; however, a recent preliminary search identified at least five primary publications assessing the psychometric properties of the STS tests since this review was published [12–16]. Additionally, since this publication [11], new guidelines have been published to direct the conduct of a systematic review of measurement properties [17]. As such, the aim of this review is to systematically review the literature to synthesize the measurement properties of the STS test variations among people with COPD.

### Research questions

What are the measurement properties (reliability, validity, and responsiveness) of the variations of the STS test in people with chronic obstructive pulmonary disease (primary)? What

are the interpretability (e.g., minimal important change, floor or ceiling effects) and feasibility characteristics (e.g., ease of administration) of the variations of the STS test (secondary)?

## Materials and methods

This systematic review protocol was registered with the International Prospective Register of Systematic Reviews (PROSPERO) on 14 September 2024 (CRD42024586635). It is reported according to the Preferred Reporting Items for Systematic Review and Meta-analysis (PRISMA) Protocols (PRISMA-P) 2015 statement (S1 Table) [18]. This review will be conducted with methods that are consistent with the updated COnsensus-based Standards for the selection of health Measurement Instruments (COSMIN) guidelines [17, 19] and reported according to the PRISMA-COSMIN 2024 statement [20].

### Eligibility criteria

Peer-reviewed publications will be eligible for inclusion if they investigated the measurement properties of a variation of the STS test in community-dwelling adults ($\geq$18 years) with COPD ($\geq$50% of the sample [17, 19] or reported as a subgroup). Measurement properties of interest include reliability (e.g., test-retest, inter-rater), validity (e.g., construct validity), responsiveness, in addition to interpretability (e.g., minimal important change) and feasibility (e.g., ease of administration) characteristics [17, 19]. Only publications with a full text available will be included [17, 19]. Publications will be excluded if participants are those with an acute exacerbation of COPD. Reports will also be excluded if the STS test was used solely as an outcome measurement instrument, was used to validate another outcome measurement instrument, was conducted remotely or virtually or if there was no aim to validate the STS test [17, 19]. Review articles will be excluded. There will be no language exclusions.

### Information sources

We will search the following databases from their inception: Medline (OVID interface, from 1946), Embase (OVID interface, from 1974), Emcare (OVID interface, from 1995), CINAHL (EBSCOhost interface, from 1981), Web of Science (Clarivate interface, from 1976) and Scopus (Elsevier interface, from 2004). We will review the reference lists of included publications to identify any other relevant reports that our database search may have missed.

### Search strategy

Search strategies will be developed in consultation with a health sciences librarian. A strategy for each database will be based on the following search concepts: (i) measurement properties, (ii) chronic obstructive pulmonary disease, and (iii) sit-to-stand test. To identify publications of measurement properties, we will use highly sensitive validated search filters, where they exist for each database [17, 19]. Search terms will include medical subject headings (MeSH) and keywords combined using Boolean operators (i.e. AND and OR) with no limits used. A preliminary search strategy for Medline is available in S2 Table.

### Screening and extraction

Citations identified from our search will be imported to EndNote 21 (2023, Clarivate available from https://endnote.com/), where duplicates will be removed. Deduplicated citations will be uploaded to Covidence systematic review software (2020, Veritas Health Innovation, Melbourne, VIC, Australia) for study selection. Two reviewers will review each title/abstract independently. If at least one reviewer deems a title/abstract potentially relevant, the full-text

publication will be sought and assessed independently and in duplicate for inclusion. Any disagreements during full-text selection will be resolved through discussion or with a third reviewer. Prior to title/abstract and full-text screening, a calibration activity of five to ten publications will be conducted to ensure consistency. Screening and selection results will be summarized in a PRISMA flow diagram.

Extraction will be completed independently and in duplicate using Microsoft Excel for Mac (2024, Microsoft Corporation available from https://office.microsoft.com/excel). The extracted information will include study characteristics (e.g., study design, geographical location, language, sample size, setting), participant characteristics (e.g., age, COPD severity), STS test characteristics (e.g., duration, repetitions), measurement properties (e.g., measurement property assessed, score, timepoint of assessment) and any descriptions of interpretability or feasibility.

## Measurement property evaluation

According to the COSMIN guidelines, each assessment of a measurement property constitutes a separate *study* [17, 19]. Thus, according to the COSMIN guidelines, each included publication may have multiple studies if it assessed multiple measurement properties (e.g., construct validity, test-retest reliability, inter-rater reliability). We will adopt this terminology throughout the description and conduct of this protocol.

Evaluation of each measurement property will occur in three stages: (1) Assessing the methodological quality of each study, (2) Rating each study against the updated criteria for good measurement properties, and (3) Rating the quality of evidence of each measurement property.

**Methodological quality.**   The methodological quality of each study will be assessed independently and in duplicate using the COSMIN Risk of Bias checklist [21]. Where disagreements exist, a third reviewer will arbitrate.

The COSMIN Risk of Bias checklist can be used to assess content validity, structural validity, internal consistency, cross-cultural validity, reliability, measurement error, criterion validity, hypothesis testing for construct validity, and responsiveness [21]. Modules for each measurement property contain three to thirty-five standards against which to rate a study's quality [21]. For each study, reviewers assign a rating ("very good," "adequate," "doubtful," or "inadequate") to all applicable measurement property standards. The study's overall rating is determined as the lowest rating given among all standards for a particular measurement property.

**Criteria for good measurement properties.**   Using the COSMIN updated criteria for good measurement properties, each study will be rated as "sufficient," "insufficient," or "indeterminant" [17, 19]. For example, for a construct validity study to be rated as sufficient, results should be in accordance with at least 75% of the hypotheses [19]. For each variation of the STS test, if the results of the previous step are consistent, they will be quantitatively pooled or narratively summarized (described in data analysis). Upon combining the individual studies, overall results for each variation of the STS test will be rated against the COSMIN updated criteria for good measurement properties ("sufficient," "insufficient," "inconsistent" or "indeterminant") [17, 19]. If results are inconsistent for a variation of the STS test, COSMIN recommends three strategies: (1) determine the reason for inconsistency and summarize by subgroup, (2) do not summarize results and do not rate the quality of evidence, or (3) report based on the consistent results and downgrade the quality of evidence rating for inconsistency. The strategy for managing inconsistency cannot be declared a priori because it will be dependent on what is found during the conduct of this review.

**Quality of evidence.**   The Grading of Recommendations Assessment, Development and Evaluation (GRADE) assesses the trustworthiness of findings across five domains: risk of bias, inconsistency, indirectness, imprecision and publication bias [22]. However, due to a lack of registries for studies of measurement properties, COSMIN recommends using a modified GRADE approach, in which publication bias is not considered [17, 19]. The quality of evidence for results deemed appropriate to pool or summarize will be assessed using this modified GRADE approach [17, 19].

## Data analysis

Categorical variables will be summarized as counts and percentages. Continuous variables will be reported as mean (standard deviation) if normally distributed and median ($1^{st}$-$3^{rd}$ quartiles) if non-normally distributed.

Results from studies that describe the same measurement property will be pooled in a meta-analysis if results are sufficiently homogeneous (i.e., same variation of the STS test, statistical approach, and comparator tool) and required data is reported (point estimate, measure of variability, sample size) [17, 19, 23]. Pooling will be conducted by calculating weighted means and 95% confidence intervals [17, 19]. Quantitative analyses will be conducted using Stata (v. 16.1 for Mac, StataCorp LP, College Station, Texas). Where studies describing the same measurement property are not sufficiently homogeneous or required data is not reported, results will be summarized narratively. Furthermore, interpretability and feasibility findings will be reported narratively.

## Dissemination of results

The results of this review will be submitted for publication in a peer-reviewed journal and for presentation at a COPD-related conference. Extracted study data will be published on Open Science Framework at the time of publication.

## Significance

Exercise intolerance is commonly experienced by people with COPD [1]. Compared to other outcome measures, such as the 6MWT, the STS tests may be more tolerable to assess exercise capacity in people with various levels of disease severity. The most recent synthesis of evidence for the STS tests in people with COPD was published in 2017 [11]. However, since then, at least five primary studies have been published [12–16]. Additionally, the COSMIN guidelines were published to inform best practices to conduct a systematic review of the measurement properties of an outcome measure [17, 19]. This review will be the first study to synthesize the measurement properties of the STS tests in people with COPD using the methods recommended by COSMIN. As such, the results of this review will be robust and may help clinicians or researchers identify the best variation of the STS test to use in their practice.

## Supporting information

**S1 Table. PRISMA-P checklist.**
(PDF)

**S2 Table. Preliminary MEDLINE search strategy.**
(PDF)

## Author Contributions

**Conceptualization:** Christopher Farley, Dina Brooks.

**Funding acquisition:** Christopher Farley.

**Methodology:** Christopher Farley.

**Project administration:** Christopher Farley.

**Supervision:** Stuart M. Phillips, Jenna Smith-Turchyn, Dina Brooks.

**Writing – original draft:** Christopher Farley.

**Writing – review & editing:** Christopher Farley, Stuart M. Phillips, Jenna Smith-Turchyn, Dina Brooks.

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
