## [Decision Letter · Decision Letter 0]

11 Dec 2024

Measurement properties of the sit-to-stand test in people with chronic obstructive pulmonary disease: Protocol for a systematic review and meta-analysis using the COSMIN guidelines

PONE-D-24-41006

Dear Dr. Farley,

We’re pleased to inform you that your manuscript has been judged scientifically suitable for publication and will be formally accepted for publication once it meets all outstanding technical requirements.

Kind regards,

Pasyodun Koralage Buddhika Mahesh

Academic Editor

PLOS ONE

Reviewers' comments:

Reviewer's Responses to Questions

**Comments to the Author**

1. Does the manuscript provide a valid rationale for the proposed study, with clearly identified and justified research questions?

Reviewer #1: Yes

Reviewer #2: Yes

2. Is the protocol technically sound and planned in a manner that will lead to a meaningful outcome and allow testing the stated hypotheses?

Reviewer #1: Yes

Reviewer #2: Yes

3. Is the methodology feasible and described in sufficient detail to allow the work to be replicable?

Reviewer #1: Yes

Reviewer #2: Yes

4. Have the authors described where all data underlying the findings will be made available when the study is complete?

Reviewer #1: Yes

Reviewer #2: Yes

5. Is the manuscript presented in an intelligible fashion and written in standard English?

Reviewer #1: Yes

Reviewer #2: Yes

6. Review Comments to the Author

You may also provide optional suggestions and comments to authors that they might find helpful in planning their study.

Reviewer #1: Introduction and the justification for the study written well and adequately. Methodology to be used is sound and feasible.

Reviewer #2: Title: Clear and well written

Abstract: Satisfactory

Main paper

Introduction: Good

Research question: Satisfactory

Materials and methods: Satisfactory

S1: Satisfactory

S2 not available

What are the main claims of the paper and how significant are they for the discipline? They have mentioned this clearly in the Significance part

Are the claims properly placed in the context of the previous literature? Yes

Have the authors treated the literature fairly? Yes

Do the data and analyses fully support the claims? Not applicable

If not, what other evidence is required? Not applicable

PLOS ONE encourages authors to publish detailed protocols and algorithms as supporting information online. Do any particular methods used in the manuscript warrant such treatment? No

If a protocol is already provided, for example for a randomized controlled trial, are there any important deviations from it? Not applicable

If so, have the authors explained adequately why the deviations occurred? Not applicable

If the paper is considered unsuitable for publication in its present form, does the study itself show sufficient potential that the authors should be encouraged to resubmit a revised version? Not applicable

Are original data deposited in appropriate repositories and accession/version numbers provided for genes, proteins, mutants, diseases, etc.? Not applicable

Does the study conform to any relevant guidelines such as CONSORT, MIAME, QUORUM, STROBE, and the Fort Lauderdale agreement? Yes

Are details of the methodology sufficient to allow the experiments to be reproduced? Yes

Is any software created by the authors freely available? Not applicable

Is the manuscript well organized and written clearly enough to be accessible to non-specialists? yes

Is it your opinion that this manuscript contains an NIH-defined experiment of Dual Use concern? Not applicable

The article adheres to appropriate reporting guidelines and community standards for data availability.

7. PLOS authors have the option to publish the peer review history of their article (what does this mean?). If published, this will include your full peer review and any attached files.

Reviewer #1: No

Reviewer #2: **Yes: **Ambepitiyawaduge Pubudu De Silva

---

## [Editor Report · Acceptance letter]

16 Dec 2024

PONE-D-24-41006 

PLOS ONE

Dear Dr. Farley, 

I'm pleased to inform you that your manuscript has been deemed suitable for publication in PLOS ONE. Congratulations! Your manuscript is now being handed over to our production team.

Kind regards, 

on behalf of

Dr. Pasyodun Koralage Buddhika Mahesh 

Academic Editor

PLOS ONE